# Tixagevimab/Cilgavimab in SARS-CoV-2 Prophylaxis and Therapy: A Comprehensive Review of Clinical Experience

**DOI:** 10.3390/v15010118

**Published:** 2022-12-30

**Authors:** Karolina Akinosoglou, Emmanouil-Angelos Rigopoulos, Georgia Kaiafa, Stylianos Daios, Eleni Karlafti, Eleftheria Ztriva, Georgios Polychronopoulos, Charalambos Gogos, Christos Savopoulos

**Affiliations:** 1Department of Internal Medicine, Medical School, University of Patras, 26504 Rio, Greece; 2First Propedeutic Department of Internal Medicine, AHEPA University Hospital, Medical School, Aristotle University of Thessaloniki, 54636 Thessaloniki, Greece

**Keywords:** COVID-19, tixagevimab/cilgavimab, Evusheld, prophylaxis, SARS-CoV-2

## Abstract

Effective treatments and vaccines against COVID-19 used in clinical practice have made a positive impact on controlling the spread of the pandemic, where they are available. Nevertheless, even if fully vaccinated, immunocompromised patients still remain at high risk of adverse outcomes. This has driven the largely expanding field of monoclonal antibodies, with variable results. Tixagevimab/Cilgavimab (AZD7442), a long-acting antibody combination that inhibits the attachment of the SARS-CoV-2 spike protein to the surface of cells, has proved promising in reducing the incidence of symptomatic COVID-19 or death in high-risk individuals without major adverse events when given as prophylaxis, as well as early treatment. Real-world data confirm the antibody combination’s prophylaxis efficacy in lowering the incidence, hospitalization, and mortality associated with COVID-19 in solid organ transplant recipients, patients with immune-mediated inflammatory diseases and hematological malignancies, and patients in B-cell-depleting therapies. Data suggest a difference in neutralization efficiency between the SARS-CoV-2 subtypes in favor of the BA.2 over the BA.1. In treating COVID-19, AZD7442 showed a significant reduction in severe COVID-19 cases and mortality when given early in the course of disease, and within 5 days of symptom onset, without being associated with severe adverse events, even when it is used in addition to standard care. The possibility of the development of spike-protein mutations that resist monoclonal antibodies has been reported; therefore, increased vigilance is required in view of the evolving variants. AZD7442 may be a powerful ally in preventing COVID-19 and the mortality associated with it in high-risk individuals. Further research is required to include more high-risk groups and assess the concerns limiting its use, along the SARS-CoV-2 evolutionary trajectory.

## 1. Introduction

Severe acute respiratory syndrome coronavirus 2, most commonly known as COVID-19 or SARS-CoV-2, is an RNA virus first detected in Wuhan, China on December 2019, its rapid transmission leading to the COVID-19 pandemic. In the following 3 years, coronavirus disease not only resulted in significant mortality and morbidity, especially in people with chronic conditions and multiple comorbidities, but also imposed a severe financial burden on health systems worldwide. Clinical manifestations of the disease are mainly driven by its biphasic nature, including an initial viral replication and toxicity stage, followed by a secondary phase of inflammatory response [1]. In the immunocompetent patient, the secondary phase is responsible for complicated disease, as reflected by severe pneumonia, the need for hospitalization, respiratory failure, the need for mechanical ventilation, intensive care unit (ICU) intake, and death. In immunocompromised patients, the inflammatory response is not so evident; prolonged viral shedding, multiple co-morbidities, and the inability to mount adequate immune response drive worse outcomes.

Exposure to SARS-CoV-2 occurs via liquid droplets and aerosol particles [2]. The upper respiratory tract, conjunctiva, and gastrointestinal tract are likely portals of entry for SARS-CoV-2 [2,3]. The highest risk of transmission occurs in the early phase of the disease, prior to experiencing symptoms [2,4], when the incubation period is estimated to be three to five days, depending on the protein S variations and virus mutations [5]. The viral load peaks within one week of symptom onset [2,4], while viral dynamics are associated with disease severity; hence, prompt intervention and therapeutic approaches are pivotal to ensure the best outcomes [2]. Although all fully vaccinated people remain at risk of contracting COVID-19, the immunocompromised remain at greater risk of adverse outcomes [6]. They are three times more likely to require hospitalization, one and a half times more likely to need ICU care, twice as likely to need vasopressor support, and twice as likely to die [7]. As a result, the need for and the possibility of pre-exposure prophylaxis in the form of monoclonal antibodies has been explored.

A number of monoclonal antibodies, including combination regimens such as Sotrovimab, Casirivimab/Imdevimab, Bamlanivimab/Etesevimab, Bebtelovimab, and Tixagevimab/Cilgavimab have been tested in clinical trials and used in clinical practice, according to the available guidelines [8]. However, the rise of mutations equally raised concerns [9] in the context of immunological escape phenomena [10], abandoning most of them along with SARS-CoV-2 evolution [8]. At the moment, genomic epidemiology has identified Omicron as the only variant of concern. In this context, the susceptibility of circulating SARS-CoV-2 variants i.e., Omicron BA.2, BA.4, BA.5, and its subvariants, i.e., BA.2.75.2, BA.4.6, and BQ.1.1, against anti-SARS-CoV-2 antibodies remains variable [8,11,12,13,14,15,16,17,18,19] (see Table 1). Sotrovimab [14,15,16,17], Casirivimab/Imdevimab [18], and Bamlanivimab/Etesevimab [18,19], although effective against the previous variants, i.e., alpha (B.1.1.7), beta (B.1.351), gamma (P.1), delta (B.1.617.2, non-AY.1/AY.2), and Omicron (B.1.1.529/BA.1 and BA.1.1), they seem to be inactive against the present ones (BA.2, BA.4, BA.5). On the contrary, Bebtelovimab and Tixagevimab/Cilgavimab, remain active in vitro against the circulating subtypes (BA.2, BA.4, BA.5) and are expected to retain their clinical activity in the future, even though the duration of their activity remains poorly defined in some cases [11,20,21]. The scientific community expressed fresh uneasiness following the emergence of new subvariants (BA.2.75.2 and BQ.1.1) evading the current antibodies, although interestingly, older regimens, e.g., sotrovimab, exhibited mild activity against BQ.1.1 [8,11,12,20,21,22].

In December 2021, the US Food and Drug Administration (FDA) issued an Emergency Use Authorization (EUA) for Tixagevimab/Cilgavimab use in individuals of 12 years and older, who weigh at least 40 kg, have moderately to severely compromised immunity, or those for whom vaccination is not recommended, due to a history of severe adverse effects from prior vaccinations. The later surge in cases, driven by Omicron subvariants BA.1 and BA.1.1, was found to have decreased COVID-19 neutralization in response to Tixagevimab/Cilgavimab [21]. As compared to the delta variant, neutralizing titers were more markedly decreased against BA.1 (344-fold) than BA.2 (nine-fold) [22]. Tixagevimab/Cilgavimab or the Tixagevimab/Cilgavimab + Casirivimab/Imdevimab neutralized the delta variant, barely neutralized BA.1, and efficiently neutralized BA.2—with only a nine-fold and 38-fold decrease in neutralization, respectively, vs. delta [23]. Thus, in February 2022, the FDA revised the EUA to include an increase in the recommended dose from 150/150 mg to 300/300 mg, based on data suggesting that the higher dose would be more likely to prevent infection by these variants [24,25]. The currently recommended dosing scheme calls for consecutive injections once every six months, if ongoing protection is needed, while SARS-CoV-2 remains in circulation [25]. Tixagevimab/Cilgavimab is the only drug indicated for the pre-exposure prophylaxis of COVID-19 in adults and adolescents, according to the Infectious Disease Society of America [25]. Tixagevimab/Cilgavimab is also indicated for the treatment of individuals with COVID-19 who do not require supplemental oxygen and who are at an increased risk of progressing to severe disease [26,27,28,29]. We aim to review the current clinical data and future perspectives regarding activity and the use of Tixagevimab/Cilgavimab.

## 2. Methods

Broad searches of Pubmed, Scopus, and Embase between 1 February 2020 and 10 December 2022 were conducted using the following keywords: ‘Tixagevimab/Cilgavimab’, “Evusheld”, “AZD7442”, “Tixagevimab” and “Cilgavimab”. Relevant publications were identified based on the titles and abstracts. No restrictions on the type of paper were set. Only English language papers were included in this review and the main focus was put on the clinical data. Three reviewers independently screened all titles and abstracts and hand-searched the references of the retrieved articles. Duplicates and irrelevant articles were removed, and all disagreements were discussed and resolved (Figure 1).

## 3. Tixagevimab/Cilgavimab

Tixagevimab/Cilgavimab (AZD7442) was the first long-acting antibody combination to demonstrate benefit in both prophylaxis and treatment. It consists of two human mAbs binding to two distinct epitopes, inhibiting attachment of the SARS-CoV-2 spike protein to the surface of cells, thereby preventing viral entry and infection by the SARS-CoV-2 virus [29,30]. The two monoclonal antibodies demonstrate the synergistic neutralization of SARS-CoV-2 in vitro [29,30], with an overall synergy δ-score of 17.4. By using a cocktail of antibodies, the dose of each mAb can be reduced by >3-fold to achieve the same potency of virus neutralization [29,30]. This is important for dose-sparing considerations throughout the pandemic [31].

The combination is quite potent [31], variably neutralizing the different variants of concern including the Omicron subvariants [9] (Table 1), and it also retains its activity against upcoming sub-lineages [23,32]. It has an extended half-life and it is administered as a single 300 mg intramuscular administration (prophylaxis dose) or 600 mg (therapeutic dose) regimen, rapidly achieving peak serum concentrations [32]. The mean elimination half-lives of Tixagevimab and Cilgavimab are 87.9 and 82.9 days, and the time to peak drug concentration is 14.9 (range 1.1–86) and 15 (range 1.1–85) days, respectively. This is as a result of antibody optimization following Fc fragment amino acid substitutions, as per M252Y, S254T, and T256E. Modified antibodies have a ~9-fold greater affinity for FcRn at pH 6.0, compared with those lacking the latter modification [33]. This promotes, on the one hand, antibody recycling, leading to an increased half-life by 4 × (20 vs. 70–100 days) [33,34,35], while, on the other hand, it maximizes antibody localization to the mucosa, resulting in significant transcytosis into the upper respiratory tract.

On top of that, the Fc segment has been triply modified (L234F, L235E, and P331S) so that the Fc effector function is ablated, reducing the binding to FcgR and C1q complement proteins, thereby decreasing the likelihood of immunopathology [33]. Neither Tixagevimab nor Cilgavimab are renally excreted or metabolized by the cytochrome P450 (CYP) enzymes; therefore, interactions with biological and medicinal products that are renally excreted or that are substrates, inducers, or inhibitors of CYP enzymes are unlikely. This is particularly important in immunocompromised patients with multiple co-morbidities who are characterized by polypharmacy. No dose adjustment is required in patients with renal or hepatic impairment.

## 4. Prophylaxis

### 4.1. Clinical Trial Data

The efficacy of a single intramuscular dose of AZD7442 for the prevention of COVID-19 prior to day 183 has been previously tested in a Phase III double-blind, placebo-controlled study for pre-exposure prophylaxis in 5197 participants, during the delta variant surge [35]. Adults aged >18 years, who are at increased risk of an inadequate response to active immunization, who would benefit from passive immunization, and who are at increased risk for SARS-CoV-2 infection were included in this study [35]. Almost half of the patients were >60 years old (43.4%). Patients suffered from multiple co-morbidities, including obesity (49.6%), cardiovascular disease (8.1%), immunosuppression (3.8%), chronic obstructive pulmonary disease (5.3%), chronic liver disease (4.6%), and chronic kidney disease (5.2%), all stable in terms of their underlying medical condition [35]. A significant reduction (RRR:83% 95% CI: 66, 91; *p* < 0.001) in the incidence of symptomatic COVID-19 with AZD7442 (0.3%) was noted compared to the placebo (1.8%) during a six-month follow-up. The time to first SARS-CoV-2 RT-PCR–positive symptomatic illness was longer for AZD7442, compared with the placebo in the 180-day follow-up. The reduction in the incidence of symptomatic COVID-19 with AZD7442, compared to the placebo, was also noted in all participants, regardless of unblinding or vaccination. Similarly, the reduction in the incidence of symptomatic COVID-19 or death due to any cause with AZD7442 compared to the placebo was recorded [35]. The efficacy of AZD7442 against symptomatic COVID-19 was generally consistent across subgroups. Adverse events (AEs) were balanced between the AZD7442 and placebo groups, while most AEs were of mild or moderate severity, including hypersensitivity reactions, clinically significant bleeding disorders, cardiovascular events, or musculoskeletal and connective tissue disorders [36,37]. Cardiovascular events were associated with previous medical history in all patients [35], even though data from a recent population-based propensity-matched cohort did not reveal any increased cardiovascular risk, even in this group of patients [36]. Nonetheless, the data still remain controversial, since a recent metanalysis showed an increased risk of cardiac and vascular adverse events [37,38]. There were two COVID-19–related deaths in the placebo group, but no COVID-19-related deaths occurred in the AZD7442 group [35]. Participants who were unblinded to receive >1 dose of the COVID-19 vaccine (AZD 1222) during the study were analyzed for vaccine-induced spike-specific IgG antibodies. No effect of AZD7442 on antibody responses following AZD 1222 COVID-19 vaccinations in humans was observed [39,40].

The assessment of Tixagevimab/Cilgavimab as post-exposure prophylaxis was attempted in the STORM-CHASER randomized clinical trial of 150 + 150 mg intramuscular administration [41]. A relative risk reduction of 33% was reported (statistically not significant) regarding symptomatic COVID-19 in the overall study population. The failure was mostly driven by the counting of cases occurring less than 7 days since administration but was also driven by PCR-positive recipients at baseline (technically no longer post-exposure prophylaxis) [41].

### 4.2. Real-World Evidence on Pre-Exposure Prophylaxis

Real-world evidence (RWE) consistently demonstrates the benefit of Tixagevimab/Cilgavimab in data from immunocompromised populations (see Table 2) [23,24,42,43,44,45,46,47,48,49,50,51,52,53,54,55,56,57,58,59,60,61,62,63,64]. Pre-exposure prophylaxis with Tixagevimab/Cilgavimab was associated with a lower risk of SARS-CoV-2 infections and better outcomes during the Omicron surge in the real world, most data deriving from the BA.2 lineage. Across three large studies [42,46,47] in a heterogeneous population of heavily immunocompromised individuals, >95% of whom were fully vaccinated, Tixagevimab/Cilgavimab reduced the risk of hospitalization or death by 80–92% [42,46,47]. In a large US study involving 1295 immunocompromised patients, 223 of whom developed COVID-19, which was also carried out during the BA.4/5 surge, lower numbers of patients developed COVID-19 before (54.3%) and after (45.7%) Tixagevimab/Cilgavimab administration [57]. Although COVID-19 cases increased with BA.2.12 and BA.5, the rate of hospitalizations did not increase, indicating the respective impact on disease severity [57].

In another three large-scale studies of solid organ transplant recipients (SOTRs) [43,52,58], of 967 SOTRs receiving Tixagevimab/Cilgavimab, >97% had been vaccinated with ≥2 COVID-19 vaccine doses; hospitalizations were reduced by 53–89% [43,52,58]. A reduction in COVID-19 breakthrough infections and severe outcomes, including ICU admission and death, in SOTRs compared with vaccinated non-responders was also noted [58]. AZD7442 (vs. the control group) was associated with a lower incidence of SARS-CoV-2 infection (HR = 0.34; 95% CI, 0.13–0.87), COVID-19 hospitalization (HR = 0.13; 95% CI, 0.02–0.99), and all-cause mortality (HR = 0.36; 95% CI, 0.18–0.73). Moreover, Tixagevimab/Cilgavimab use was safe and was associated with a lower risk of breakthrough SARS-CoV-2 infection in 222 vaccinated SOTRs during the Omicron wave [43]. At a mean follow-up of 67 ± 18 days after Tixagevimab/Cilgavimab administration, breakthrough infections occurred in 1.8% of Tixagevimab/Cilgavimab group (one hospitalization, zero deaths) versus 4.7% in the vaccinated control group (five hospitalizations, four deaths) [43]. AEs occurred in nine SOTRs at a median of 15 days after Tixagevimab/Cilgavimab administration [43]. However, BA.1 and BA.2 exhibit noticeable differences in their sensitivity to therapeutic mAbs. In a prospective observational study, Tixagevimab/Cilgavimab was evaluated as pre-exposure prophylaxis in SOTRs in a real-world setting during the BA.1/BA.2 period [45]. BA.2 neutralization increased from 7% to 72% of participants post-Tixagevimab/Cilgavimab (*p* < 0.001). Tixagevimab/Cilgavimab increased the anti-RBD levels, yet BA.1-neutralizing activity was minimal. The incidence of breakthrough infections was lower among patients treated with Tixagevimab/Cilgavimab, especially when the BA.2 sublineage was predominant [48]. The viral-neutralizing activity of the serum against the BA.1 variant has been disappointing among kidney transplant recipients [59,60]. No significant changes were observed in the serum creatinine or liver laboratory values in kidney or liver transplant recipients, respectively [43].

In accordance with these findings, experience from 463 immunocompromised patients, including 18% following a solid organ or hematopoietic stem cell transplantation, who were administered Tixagevimab/Cilgavimab as pre-exposure prophylaxis, showed that only 42 patients (9.1%) were hospitalized, and 4 (0.9%) died, but none was attributed to COVID-19 [53]. The median days from Tixagevimab/Cilgavimab administration to non-COVID-19-related hospitalization and death were 30 (IQR 17, 55) and 53 (IQR 18, 91), respectively [53]. Similarly, in a series of 161 recipients of allogeneic hematopoietic stem-cell transplants receiving Tixagevimab/Cilgavimab, 86.3% remained uninfected, while no hospitalizations or deaths were reported [61].

A significant benefit is also demonstrated in patients with hematological malignancies [23]. All 52 patients who were administrated Tixagevimab/Cilgavimab, with 92% of whom having received at least one dose of vaccination, achieved high anti–SARS-CoV-2 antibody titers [23]. Through a median follow-up of 79 days, 96% of patients did not have a documented SARS-CoV-2 infection, while everyone recovered without hospitalization or death [23]. However, data demonstrated the remaining risk of breakthrough COVID-19 infection in patients with B-cell malignancies who received pre-exposure prophylaxis with Tixagevimab/Cilgavimab and B-cell-depleting therapy or hematological stem cell recipients within the past three to six months [23,54].

A larger study of immunocompromised patients, with a median follow-up of 63 days, found a breakthrough infection rate of 4.4%, although the majority of the included patients did not have a hematological malignancy [47]. In another retrospective analysis of immunocompromised patients who were given Tixagevimab/Cilgavimab as pre-exposure prophylaxis while Omicron VOC was dominant, only 1.2% of patients developed COVID-19, 87.5% of whom had previously been vaccinated [48]. Even though these outcomes were shown prior to the increased 300/300 dose recommended by the FDA, caution should be shown in certain subpopulations of hematological patients, even with increased dose administration in the context of BA.5 [54,62].

Data extends beyond the hematological departments. Experiences from 412 patients with immune-mediated inflammatory diseases and those with inborn errors of humoral immunity across the rheumatology [45,64], allergy [45], and neurology departments [45,65] came to show that Tixagevimab/Cilgavimab prevented hospitalizations in 91.7% of patients with COVID-19. Only 12 patients (2.9%) experienced a breakthrough COVID-19 infection, all being treated with B-cell-depleting therapies. Six patients developed an infection at a median of 19 days (13–84) after receiving 150 mg/150 mg of Tixagevimab/Cilgavimab, while another 6 patients developed an infection at a median of 38.5 days (19–72) after either a single dose of 300 mg/300 mg or after their second dose of 150 mg/150 mg [45]. All patients except one recovered at home, following a mild course of treatment, while there were no deaths reported. Notably, all cases had been previously vaccinated against COVID-19 [45]. A similar benefit was shown in rheumatologic patients on rituximab [51,64], where Tixagevimab/Cilgavimab prevented hospitalizations by 100%. After regimen receipt (300 mg, 600 mg), only 1.2% of patients developed COVID-19, 98% did not develop symptomatic COVID-19, and 100% recovered without additional treatment, while a 49% reduction in SARS-CoV-2 incidence rate vs. the local population rate was noted [51]. In a retrospective cohort study of 1848 patients deriving from the US Department of Veteran Affairs, who were treated with at least one dose of intramuscular Tixagevimab/Cilgavimab, 69% of whom were ≥65 y/o, Tixagevimab/Cilgavimab use prevented composite COVID-19 outcomes in 67% of older participants [59]. Of note, the study population had a high prevalence of comorbidities, e.g., hypertension (59%), diabetes (31%), cancer (34%), renal disease (25%), etc., while 92% were immunocompromised [59].

In a propensity-matched analysis of 703 immunocompromised patients in Israel, HR was 0.75 (95% CI, 0.58–0.96) in terms of SARS-CoV-2 infection, and 0.41 (0.19–0.89) for COVID-19-related hospitalization in the Tixagevimab/Cilgavimab group compared to the control group [55]. The magnitude of the relative risk reduction of each outcome was greater in non-obese patients (*p* = 0.020 and 0.045, respectively) [55]. The magnitude of the SARS-CoV-2 infection risk reduction associated with Tixagevimab/Cilgavimab was lower in the RWE compared to the PROVENT trial, with an HR of 0.75 [55] and 0.17 [36], respectively, similar to what has been noted with nirmatrelvir/ritonavir [65]. This could be attributed to a number of reasons. While the PROVENT trial included a low percentage of immunosuppressed patients (<5%) and the study population included, among others, healthy individuals who are at risk of SARS-CoV-2 exposure, and only those unvaccinated patients, Najjar-Debiny’s study included solely immunocompromised patients, regardless of their vaccination status. The outcome investigated in the PROVENT study was also on symptomatic COVID-19 infection, while RWE investigated the outcome of a laboratory-confirmed COVID-19 infection, regardless of symptoms. Finally, the PROVENT study was conducted when the alpha, beta, and delta variants were still the circulating variants, while our study was conducted when the prevailing variant in Israel was the Omicron one.

## 5. Therapy

### 5.1. Clinical Trial Data

Following SARS-CoV-2 infection, approximately 81% of patients will develop a mild disease that, in the majority of cases, will end in symptom resolution and full recovery, while in 33%, at least one persistent symptom will occur, constituting long-COVID syndrome [66,67]. In 14% of infected patients, however, with predisposing factors, including an age > 60 years, increased BMI, chronic co-morbidities, immunosuppression, etc., clinical deterioration is common, leading to ICU admission and death, comprising an overall COVID-19 case-fatality rate of 2.3% and a hospital case-fatality rate of 13.6% [68].

TACKLE was a phase-III double-blind, randomized, placebo-controlled study of AZD7442 for the treatment of COVID-19 in 903 outpatient adults at high risk of progression to severe COVID-19 [51]. The majority of individuals were <60 years of age and 60% of individuals were treated within 5 days of symptom onset. Investigators observed a statistically significant reduction (RRR: 50.49% (95% CI: 14.56, 71.31; *p* < 0.010)) in severe COVID-19 cases or death with AZD7442 (4.4%), compared with the placebo (8.9%) [51]. With each additional day after symptom onset, the efficacy decreases, on average, by approximately 10%, which is in line with the biphasic nature of the disease, requiring antiviral intervention early in the course of the infection.

Similarly, in another phase III trial (ACTIV-3), the investigators evaluated a single intravenous dose of Tixagevimab/Cilgavimab, in addition to remdesivir and other standard care [69]. Tixagevimab/Cilgavimab did not improve the primary endpoint of sustained patient recovery, but it was safe and led to a clinically relevant reduction in mortality [69]. The mortality signal was numerically larger in patients requiring high-flow oxygen or non-invasive mechanical ventilation at study entry and in patients infected with the delta SARS-CoV-2 variant [69].

### 5.2. RWE on Therapy

At the time that this review was written, Tixagevimab/Cilgavimab had received EMA approval as an indicated early therapy for COVID-19, at a dosage scheme of 300 mg + 300 mg, but not as yet from the FDA. Hence, to the best knowledge of these authors, the RWE of Tixagevimab/Cilgavimab pertaining to early therapy is limited [70,71,72]. Experience in 13 patients with hematological malignancies during the Omicron surge, of whom 9 were vaccinated, full recovery was recorded in 4/8 patients that presented with a symptomatic disease requiring supplemental oxygen, following Tixagevimab/Cilgavimab administration. Early therapy in 61 kidney-transplant recipients at high risk, i.e., aged >60 years, with diabetes, obesity, or cardiovascular disease, resulted in significantly less frequent COVID-19-related hospitalizations (3.8% versus 34%, *p* = 0.006) and the need for oxygen (3.8% versus 23%, *p* = 0.04), as well as non-significant trends toward a lower number of ICU admissions (3.8% versus 14.3% *p* = 0.17) and deaths (0 versus 3, *p* = 0.13). However, no major benefit was noted in low-risk patients [70]. In another, single-center retrospective case series of immunocompromised patients, 13 of whom received Tixagevimab/Cilgavimab as a targeted treatment, no one died, even though 42 and 17% were in need of hospitalization and ICU care, respectively [71].

Recent meta-analysis confirmed the favorable results in a heterogenous mixture of studies, including both pre-exposure prophylaxis and treatment, showing that the overall mortality rate in the Tixagevimab/Cilgavimab-treated group was significantly lower than that in the control group (RR = 0.50, 95% CI: 0.39, 0.64, *p* < 0.01) [73]. In addition, protection against COVID-19 was significantly improved in the Tixagevimab/Cilgavimab group compared with the control group (RR = 0.28, 95% CI: 0.15, 0.53, *p* < 0.01) [73]. Tixagevimab/Cilgavimab treatment was not associated with the development of serious adverse events in patients (OR = 0.90, 95% CI: 0.67, 1.21, *p* = 0.48; I 2 = 0%).

## 6. Expert Opinion and Future Directions

Tixagevimab/Cilgavimab appears to be an effective and safe option for treatment among immune-compromised patients who are unable to mount an adequate immunologic response following vaccination and, hence, remain at high risk of COVID-19 severe disease. Nonetheless, there is still controversy remaining around the subgroup of patients that the definition of immunocompromise includes. At the moment, hematological patients have been set as a priority; however, patients with other malignant diseases, those undergoing chemotherapy, patients undergoing renal replacement therapy, patients undergoing monoclonal antibodies for autoinflammatory diseases, etc., are among many that could be majorly benefited if a broader application of Tixagevimab/Cilgavimab is to be considered. The high acquisition cost could be significantly outweighed by the cost attributed to prolonged hospitalization, the need for intensive care, or work-absence days. Moreover, the availability of drugs is also hampered due to a lack of universal approval by the regulatory authorities within Europe [74]. Marketing authorization within the EU followed that in the US, in March 2022, while different dosage schemes of 150/150 mg as pre-exposure prophylaxis and 300/300 mg given only as a therapy are valid, where they are applicable throughout the continent. Similarly, logistic details regarding the availability of infusion referral centers or facilities to allow the safe and timely administration of the drugs under close monitoring shortly after infusion remain of concern [75]. Finally, on top of the lack of feasibility of performing timely screening for baseline antibodies comes the absence of a clear titer cut-off of humoral and/or cellular response as to what represents adequate immunity.

Constant monitoring of the circulation of the variants of concern, in order to select the appropriate treatment or to exclude patients for whom the administration of mAb may be ineffective, is necessary. Recent reports have highlighted the risk of developing spike-protein mutations that confer resistance to Cilgavimab in patients previously given Tixagevimab/Cilgavimab, as occurred in patients treated with sotrovimab alone [76,77]. Especially among immunocompromised patients, in the context of new Omicron subvariants BA.4/5 against which the neutralizing activity of Cilgavimab is lower than that against BA.2, close virological monitoring is pivotal to minimize the risk of transmission of resistant variants in the community, setting the recent recommendations for increased dosage. Currently, emerging subvariants, including BA.2.75.2 and BQ.1.1, raise significant concerns since both Casirivimab/Imdevimab and Tixagevimab/Cilgavimab appear to have lost their antiviral properties [11]. Most recently, the rapidly increasing BQ.1.1 subvariant, reaching approximately 40% in the US, forced the National Institutes of Health COVID-19 treatment guidelines’ recommendation panel to abandon the use of bebtelovimab as an early therapy [8]. Interestingly, though, sotrovimab was found to retain mild activity against BQ.1.1, similar to sera from patients following BA.5 breakthrough infections [11]. At the time that this review was written, the panel recognized that Tixagevimab/Cilgavimab remains the only agent approved for pre-exposure prophylaxis in patients unable to mount an adequate immune response and continue to recommend its use; nonetheless, living guidelines may change in view of evolving resistance. In this context, attempts to investigate the two ingredients (Tixagevimab or Cilgavimab), both individually in increased dosages and in alternate schemes, given the likelihood of resistance emerging along the Omicron evolution trajectory, are reasonable. The Astra-Zeneca-sponsored NCT05166421 is investigating individual ingredients versus the cocktail given as pre-exposure prophylaxis in adults > 18 years [78]. Due to the rapidly evolving field of existing mutations and in view of their persisting potency, the scientific community has, at the time of writing, prioritized the use of antivirals over monoclonals in COVID-19 management, while a potential combination use remains under discussion [8,9].

The safety of the use of anti-SARS-CoV-2 mAbs for pregnant women is not well defined; however, these treatments appear to reduce the risk of a severe disease without increasing the risk of significant adverse maternal or perinatal outcomes [79,80]. More trials are ongoing in pediatric patients. The phase-I NCT05281601 study will investigate the safety of intramuscular or intravenous Tixagevimab/Cilgavimab in patients aged >29 weeks of gestational age to <18 years, while the phase-II NCT05375760 (ENDURE) study will investigate the pre-exposure prophylaxis, in moderately to severely immunocompromised patients aged >12 years, with Tixagevimab/Cilgavimab (300 + 300 mg) intramuscularly, every 3 versus every 6 months [78].

More observational studies in pre-exposure prophylaxis are currently in progress, including in immunocompromised patients, including COVIMAB in France (NCT05439044), PREP in the USA (NCT05461378), and PRECOVIM in France (NCT05216588), but also in specific patient subgroups (e.g., NCT05438498 in cancer patients in the USA or TIXCI-TRANS in SOTRS in France (NCT05234398). At the moment, we are looking forward to EVOLVE (NCT05315323), a multi-country, multi-center, single-arm, observational, prospective study using primary data collection to describe the demographic and clinical characteristics of patients who received the first dose of AZD7442 for the prevention of SARS-CoV-2 infection causing symptomatic COVID-19 illness.

## 7. Limitations

This was a literature review of data regarding patients’ experience with Tixagevimab/Cilgavimab use in the clinical setting throughout the last two years of the SARS-CoV-2 pandemic. Even though the field remains limited in comparison to other prophylaxis and early therapy regimens, Tixagevimab/Cilgavimab experience is rapidly evolving in the context of an existing pandemic and an immunocompromised population; hence, the authors run the risk of being out of date by the time that this review is published. Moreover, we included only English-language papers in our search. It is possible that experiences recorded in languages other than English are subject to selection and publication bias and were, thus, not included in this review.

## 8. Conclusions

In conclusion, in the setting of the current pandemic, Tixagevimab/Cilgavimab represents a useful tool for patients not able to mount an adequate response. However, in the setting of evolving mutations, increased surveillance, and genomic testing is pivotal, where available. In the case of susceptibility, the efficacy results and safety profile of Tixagevimab/Cilgavimab allow for a discussion of its broader use, beyond strictly severely immunocompromised patients. More trials are necessary to determine the optimal dosage or to provide for a constantly adjusting molecule that can overcome the evolving variants’ resistance. Physicians and patients should be aware of their options and should carefully adopt their strategy on a case-by-case basis.

## Figures and Tables

**Figure 1 viruses-15-00118-f001:**
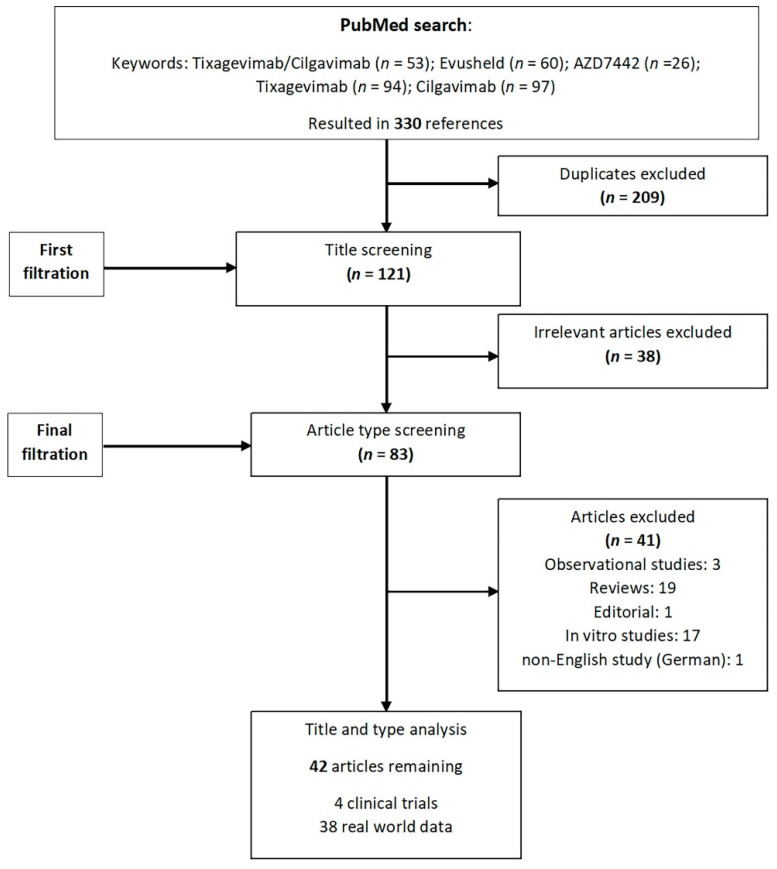
Flowchart of included and excluded studies.

**Table 1 viruses-15-00118-t001:** Monoclonal antibodies approved for clinical use and the SARS-CoV-2 variant evasion of neutralization.

Lineage	Tixagevimab/Cilgavimab	Sotrovimab (no Longer Recommended, According to NIH Living Guidance)	Casirivimab/ Imdevimab (no Longer Recommended, According to NIH Living Guidance)	Bebtelovimab (no Longer Recommended, According to NIH Living Guidance)	Bamlanivimab/ Etesevimab (no Longer Recommended, According to NIH Living Guidance)
Alpha					
Beta					
Gamma					
Delta					
Omicron					
BA.1					
BA.1.1					
BA.2					
BA.2.12.1					
BA.2.75.2					
BA.4					
BA.4.6					
BA.5					
BQ.1/BQ1.1					
XBB (BA2.10.1 and BA.2.75 recombinant)					

Green, orange and red indicate no, mild/moderate, and severe reduction in potency in vitro. Data composition as per [8,9,10,11,12,13,14,15,16,17,18,19].

**Table 2 viruses-15-00118-t002:** Real-world evidence on the effectiveness of Tixagevimab/Cilgavimab in terms of pre-exposure prophylaxis.

Reference	Country of Origin	Population Characteristics	Outcomes
Pre-exposure prophylaxis			
Young-Xu, et al. Tixagevimab/Cilgavimab for Prevention of COVID-19 during the Omicron Surge: Retrospective Analysis of National VA Electronic Data. medRxiv 2022. **[42]** https://doi.org/10.1101/2022.05.28.22275716 (accessed on 10 December 2022)	USA	Adults (69% >65 years old) Immunocompromised 73% vaccinated N(Tixa/Cilga) 1733 N(control) 6354	**Composite endpoints: SARS-CoV-2 infection, COVID-19-related hospitalization, all-cause mortality** Lower incidence of composite outcome 17/1733 (1.0%) vs 206/6354 (3.2%); HR 0.31; 95% CI, 0.18–0.53) lower SARS-CoV-2 infection (HR 0.34; 95% CI, 0.13–0.87) lower COVID-19 hospitalization (HR 0.13; 95% CI, 0.02–0.99) lower all-cause mortality (HR 0.36; 95% CI, 0.18–0.73)
Ordaya EE, et al. Characterization of Early-Onse Severe Acute Respiratory Syndrome Coronavirus 2 Infection in Immunocompromised Patients Who Received Tixagevimab-Cilgavimab Prophylaxis *Open Forum Infect Dis*. 2022. **[48]** https://doi.org/10.1093/ofid/ofac283 (accessed on 10 December 2022)	USA	Adults Immunocompromised N(Tixa/Cilga) 674 N(control) 1080	**Endpoint: SARS-CoV-2 infection, COVID-19-related hospitalization, all-cause mortality** 8/674 (1.2%) infected with COVID-19 2/8 required hospitalization No deaths
Kertes J et al. Association Between AZD7442 (Tixagevimab-Cilgavimab) Administration and Severe Acute Respiratory Syndrome Coronavirus 2 (SARS-CoV-2) Infection, Hospitalization, and Mortality, *Clinical Infectious Diseases*, 2022; ciac625, **[46]** https://doi.org/10.1093/cid/ciac625 (accessed on 10 December 2022)	Israel	Adults Immunocompromised N(Tixa/Cilga) 825 N(control) 4299	**Endpoint: SARS-CoV-2 infection, COVID-19-related hospitalization, all-cause mortality** 29/825 (3.5%) and 308/4299 (7.2%) infected with COVID-19 1/825(0.1%) compared with 27/4299(0.6%) hospitalized 0/825 compared with 40/4299(0.9%) mortality rate The AZD7442 group was half as likely to become infected with SARS-CoV-2 (OR: 0.51; 95% CI: 0.30–0.84) 92% less likely to be hospitalized/die than those not administered AZD7442 (OR: 0.08; 95% CI: 0.01–0.54).
Stuver, R. et al. Activity of AZD7442 (tixagevimab-cilgavimab) against Omicron SARS-CoV-2 in patients with hematologic malignancies, *Cancer Cell* 2022, 40, 590–591. **[23]** https://doi.org/10.1016/j.ccell.2022.05.007 (accessed on 10 December 2022)	USA	52 patients with hematologic malignancies 38.5% non-Hodgkin lymphoma 46.2% prior stem cell transplant or chimeric antigen receptor T cell therapy 47 received a 150 mg single dose, 17 received an additional 150 mg dose 5 received a 300 mg single dose	**Endpoint: anti-S IgG titers** 47/47(100%) high titers **neutralization of wild-type (WT) receptor-binding domain (RBD)** 47/47 (100%) who received single dose 150 mg 5/5(100% who received additional 150 mg 5/5(100%) who received single dose of 300 mg **neutralizing activity against Omicron-RBD (positive cut-off value = 30%)** The median neutralization by subgroup of 47 was <30%; those who received 300 mg in total had a mean neutralization >30% (9/10 above 30%) differential neutralizing capacity against various Omicron sublineages; 300 mg dose of tixa/cilga for pre-exposure prophylaxis
Benotmane, I. et al. Pre-exposure prophylaxis with Evusheld™elicits limited neutralizing activity against the Omicron variant in kidney transplant patients. medRxiv 2022. **[59]** https://doi.org/10.1101/2022.03.21.22272669 (accessed on 10 December 2022)	France	63 Kidney transplant recipients No history of COVID-19 No positive anti-nucleocapsid IgG 14 SARS-CoV-2 positive patients during the fifth wave 39 received prophylactic casirivimab-imdevimab	**Primary endpoint: Omicron BA.1 neutralization activity after 29 days (median)** 9.5% (6/63) of those who received Evusheld 71% (10/14) of those positive during the fifth wave 2.6% (1/39) of those who received casirivimab-imdevimab **Secondary endpoint: anti-RBD IgG titers** generally low after Evusheld injection high interindividual variability the patients’ body mass index has an inverse correlation with anti-RBD IgG titers no neutralizing activity with anti-RBD titers <2500 BAU/mL after Evusheld
Bertrand, D. et al. Efficacy of anti–SARS-CoV-2 monoclonal antibody prophylaxis and vaccination on the Omicron variant of COVID-19 in kidney transplant recipients. Letter. *Kidney Int*. 2022. **[58]** https://doi.org/10.1016/j.kint.2022.05.007(accessed on 10 December 2022)	France	860 Kidney transplant recipients Fully vaccinated Group 1 vaccine only: 288 Group 2 tixa/cilga: 412 (267 received casirivimab-imdevimab before tixa/cilga) Group 3 insufficient immunization: 160 (62 received casirivimab-imdevimab)	**Endpoint: Incidence of Omicron SARS-CoV-2 infection, COVID-19-related hospitalization, all-cause mortality** 113/860(13.1%) infected with COVID-19 21/860(2%) required hospitalization (8 in the ICU) 5/860 (0.6%) COVID-19-related deaths The occurrence of infection, symptomatic infection, hospitalization, intensive care unit hospitalization, and COVID-19 death were significantly increased in patients in group 3 Patients in groups 1 and 2 showed similar results
Kaminski, H. et al. COVID-19 morbidity decreases with tixagevimab–cilgavimab preexposure prophylaxis in kidney transplant recipient nonresponders or low-vaccine responders. Letter in press. *Kidney Int*. 2022. **[52]** https://doi.org/10.1016/j.kint.2022.07.008 (accessed on 10 December 2022)	France	430 kidney transplant recipients N tixa/cilga: 333 N without tixa/cilga:97	**Endpoint: Incidence of Omicron SARS-CoV-2 infection, COVID-19-related hospitalization, all-cause mortality** 41/333 (12.3%) and 42/97 (43.3%) infected with COVID-19 4/333 (1.2%) and 11/97 (11.3%) required hospitalization (2 and 6 KTR respectively required in the ICU) 1/333 (0.3%) and 2/97 (2%) COVID-19-related deaths preexposure prophylaxis with tixagevimab–cilgavimab is effective onCOVID-19 infection caused by Omicron in KTRs
Al Jurdi et al. Tixagevimab/cilgavimab pre-exposure prophylaxis is associated with lower breakthrough infection risk in vaccinated solid organ transplant recipients during the Omicron wave. Online ahead of print. *Am. J Transplant*. 2022. **[43]** https://doi.org/10.1111/ajt.17128 (accessed on 10 December 2022)	USA	Solid organ transplant recipients Group1 (tixa/cilga 150–150 or 300–300 mg dose): 222 Group 2 (vaccine only): 222	**Endpoint: Incidence of breakthrough COVID-19 infection** Breakthrough infection in 11 (5%) from group 1 and 32 (14%) from group 2 150–150 mg dose subgroup had a higher incidence of breakthrough infections compared to those who received the 300–300 mg dose
Karaba et al. Omicron BA.1 and BA.2 Neutralizing Activity following Pre-Exposure Prophylaxis with Tixagevimab plus Cilgavimab inVaccinated Solid Organ Transplant Recipients. medRxiv 2022. **[45]** https://doi.org/10.1101/2022.05.24.22275467 (accessed on 10 December 2022)	USA	61 Solid organ transplant recipients Group 1: 21 received single 300 + 300 mg dose Group 2: 40 received two 150–150 mg doses Vaccinated with at least three doses	**Endpoints: Neutralization of SARS-CoV-2 variants after tixa/cilga (achieving ≥20% ACE2 inhibition)** Omicron BA.1: from 5/61 (8%) to 10/61 (16%) (*p*-value:0.06) Omicron BA.2: from 4/61 (7%) to 44/61 (72%) (*p*-value < 0.001) The change in titer was similar for those receiving a single 300 + 300 mg dose, versus two 150 + 150 mg doses.
Conte, W.L. et al. Tixagevimab and Cilgavimab (Evusheld) boost antibody levels to SARS-CoV-2 in patients with multiple sclerosis on b-cell depleters. *Mult Scler Relat Disord.* 2022. **[49]** https://doi.org/10.1016/j.msard.2022.103905(accessed on 10 December 2022)	USA	18 MS patients on B-cell depleters Vaccinated	**Endpoints: Level of SARS-CoV-2 antibody response** At baseline 12/18 were lower than 0.8 U/mL and 6/18 were above threshold Two weeks after tixa/cilga 100% had an antibody response above threshold (>250 U/mL; *p*-value < 0.001)
Ocon, A.J. et al. Real-World experience of Tixagevimab and Cilgavimab (Evusheld) in rheumatologic patients on Rituximab **[50]** https://doi.org/10.1097/rhu.0000000000001907 (accessed on 10 December 2022)	USA	43 rheumatologic patients on Rituximab	**Endpoint: Infection with SARS-CoV-2 after 100 ± 33 days** 1/43 experienced symptomatic infection No serious adverse events occurred
Calabrese, C. et al. Early experience with tixagevimab/cilgavimab pre-exposure prophylaxis in patients with immune-mediated inflammatory disease undergoing B cell depleting therapy and those with inborn errors of humoral immunity. Letter and supplementary data. *RMD Open*. 2022;8(2):e002557 **[44]** http://dx.doi.org/10.1136/rmdopen-2022-002557 (accessed on 10 December 2022)	USA	412 patients with immune-mediated inflammatory disease on b-cell depleting therapy inborn errors of humoral immunity group 1: 150–150 mg single dose group 2: 300–300 mg single dose or second dose of 150–150 mg	**Endpoints: SARS-CoV-2 infection, COVID-19-related hospitalization, mortality** 12/412 (2.91%) developed breakthrough COVID-19 6 were from group 1 and 6 from group 2 Group 1 patients developedinfection a median of 19 days (13–84) Group 2 patients developed infection a median of 38.5 days (19–72) One patient required hospitalization and high-flow oxygen There were no deaths
Chen, B. et al. Real-World Effectiveness of Tixagevimab/cilgavimab (Evusheld) in the Omicron Era. Pre-print. medRxiv. 2022 **[57]** https://doi.org/10.1101/2022.09.16.22280034 (accessed on 10 December 2022)	USA	1295 patients who received tixagevimab/cilgavimab 37.25% were SOTRs 47.7% had received bone marrow transplants or had hematological malignancies 15.1% had other conditions (active chemotherapy, advanced HIV/AIDS, on immunosuppressants)	**Endpoints: SARS-CoV-2 infection, COVID-19-related hospitalization, mortality** SARS-CoV-2 infection: 121/1295 (9.3%) before and 102/1295 (7.9%) after receiving tixa/cilga Hospitalization: 36/121 (29.8%) (8/36 required ICU) and 6/102 (5.9%) No COVID-19-related deaths occurred
Nguyen, Y. et al. Pre-exposure prophylaxis with tixagevimab and cilgavimab (Evusheld) for COVID-19 among 1112 severely immunocompromised patients. Research note in press. *Clin Microbiol Infect*. 2022. **[47]** https://doi.org/10.1016/j.cmi.2022.07.015 (accessed on 10 December 2022)	France	1112 immunocompromised patients SOTRs 631/1112 (Kidney 511/631; Heart 83/631; Lung 36/631; Liver 1/631) Hematologic malignancies 306 Patient in need of immunosuppressants	**Endpoints: SARS-CoV-2 infection, severity of illness, mortality after median 63 (49–73) days** SARS-CoV-2 infection: 49/1112 (4.4%) ≥ 5 days after treatment Mild to moderate illness: 43/49 (88%) Moderate-to-severe illness:6/49 (12%) Deaths:2/49 (4%)
Benotmane, I. et al. Breakthrough COVID-19 cases despite prophylaxis with 150 mg of tixagevimab and 150 mg of cilgavimab in kidney transplant recipients. **[60]** https://doi.org/10.1111/ajt.17121 (accessed on 10 December 2022)	France	416 Kidney transplant recipients All received 150/50 mg single dose of tixa/cilga	**Endpoints: SARS-CoV-2 infection, COVID-19-related hospitalization, mortality** SARS-CoV-2 infection: 39/419 (9.4%) Hospitalization:14/39 (35.9%) (3 patients were admitted to the ICU) Deaths: 2/39 (5.1%) Pre-exposure prophylaxis with Evusheld™ does not adequately protect KTRs against Omicron
Al-Obaidi, M.M., Gungor A.B., Kurtin S.E., Mathias A.E., Tanriover B., and Zangeneh, T.T. The Prevention of COVID-19 in High-Risk Patients Using Tixagevimab-Cilgavimab (Evusheld): Real-World Experience at a Large Academic Center. *Am J Med*. 2022. **[53]** https://doi.org/10.1016/j.amjmed.2022.08.019 (accessed on 10 December 2022)	USA	463 immunocompromised patients (Transplant recipients, Hematologic malignancies, autoimmune disease, advanced HIV disease, on chemotherapy) 76.9% vaccinated with at least one dose for COVID-19 Total dose of 300/300 mg tixa/cilga (single 300/300 or two 150/150)	**Endpoints: SARS-CoV-2 infection, COVID-19-related hospitalization, mortality** SARS-CoV-2 infection: 6/98 (who had PCR test available) Hospitalization: 42/463 (9.1%) Deaths: 4/463(0.9%). no deaths were attributed to COVID-19
Davis, J.A., Granger, K., Roubal, K., Smith, D., Gaffney, K.J., McGann, M. et al. Efficacy of tixagevimab-cilgavimab in preventing SARS-CoV-2 for patients with B-cell malignancies. *Blood.* 2022 **[54]** https://doi.org/10.1182/blood.2022018283 (accessed on 10 December 2022)	USA	251 patients with B-cell malignancies 14/251 (5.6%) in Group 1: single dose 150/150 mg tixa/cilga 237/251 (94.4%) in Group 2: single dose 300/300 mg or two doses of 150/150 mg tixa/cilga	**Endpoints: incidence of COVID-19 breakthrough infections COVID-19-related hospitalization, mortality** Breakthrough cases at median 91-day follow-up: 27/251 (10.7%) Hospitalization: 4/27 (15%) No deaths observed
Najjar-Debbiny, R., Gronich, N., Weber, G., Stein, N., Saliba, W. Effectiveness of Evusheld in Immunocompromised Patients: Propensity Score-Matched Analysis. *Clin Infect. Dis.* 2022 **[55]** https://doi.org/10.1093/cid/ciac855 (accessed on 10 December 2022)	Israel	703 immunocompromised patients N(control): 2812	Endpoints: SARS-CoV-2 infection, COVID-19-related hospitalization SARS-CoV-2 infection: 72/703 (10.2%) and 377/2812 (13.4%); HR 0.75 (95% CI, 0.58–0.96); *p*-value: 0.023 Hospitalization: 7/72 and 67/377; HR 0.41 (0.19–0.89); *p*-value: 0.025
Zerbit, J. et al. Patients with Hematological Malignancies Treated with T-Cell or B-Cell Immunotherapy Remain at High Risk of Severe Forms of COVID-19 in the Omicron Era. *Viruses* 2022, 14, 2377. **[56]** https://doi.org/10.3390/v14112377 (accessed on 10 December 2022)	France	338 patients with hematological malignancies	**Endpoints: SARS-CoV-2 infection, COVID-19-related hospitalization, mortality** SARS-CoV-2 infection: 57/338 (16.9%) Hospitalization: 13/57 (22.8%), of whom 11/13 (84.6%) required invasive mechanical ventilation 3 deaths were recorded
Jondreville L.; et al. Pre-exposure prophylaxis with tixagevimab/cilgavimab (AZD7442) prevents severe SARS-CoV-2 infection in recipients of allogeneic hematopoietic stem cell transplantation during the Omicron wave: a multicentric retrospective study of SFGM-TC. *J Hematol Oncol.* 2022 Nov 28;15(1):169 **[61]** https://doi.org/10.1186/s13045-022-01387-0 (accessed on 10 December 2022)	France	161 recipients of allogeneic hematopoietic stem cell transplant anti-SARS-CoV-2-spike IgG titers < 260 (BAU)/mL negative test for SARS-CoV-2 117/161 (73%) four times vaccinated One dose of Tixa/cilga 150/150 mg	**Endpoints: SARS-CoV-2 infection, COVID-19-related hospitalization, mortality** 139/161 (86.3%) remained uninfected 22/161(13.7%) symptomatic SARS-CoV 2 infection 8/22 (36.4%) received an additional treatment No hospitalizations recorded No deaths recorded
Aqeel, F., and Geetha, D. (2022). Tixagevimab and Cilgavimab (Evusheld) in Rituximab-treated Antineutrophil Cytoplasmic Antibody Vasculitis Patients. *Kidney International Reports*, *7*(11), 2537–2538. **[63]** https://doi.org/10.1016/j.ekir.2022.08.019 (accessed on 10 December 2022)	USA	21 patients with antineutrophil cytoplasmic antibody-associated vasculitis treated with rituximab 20/21 received one dose of Tixa/Cilga 300/300 mg 1/21 received one dose of Tixa/Cilga 150/150 mg	**Primary Endpoints: SARS-CoV-2 infection, COVID-19-related hospitalization, mortality** **The one patient who received the lower** **Evusheld dose was infected with SARS-CoV-2 122 days after receiving Evusheld** 3/20 (15%) developed breakthrough COVID-19 disease No hospitalizations recorded No deaths recorded
Woopen, C., Konofalska, U., Akgün, K., and Ziemssen, T. (2022). Case report: Variant-specific pre-exposure prophylaxis of SARS-CoV-2 infection in multiple sclerosis patients lacking vaccination responses (Case Report). *Frontiers in Immunology*, 13. **[64]** https://doi.org/10.3389/fimmu.2022.897748 (accessed on 10 December 2022)	Germany	6 patients with multiple sclerosis on treatment with sphingosine-1-phosphate receptor modulators Failed to develop SARS-CoV-2-specific antibodies and T-cells after three vaccinations Initial treatment with casirivimab/imdevimab in times of a predominance of the SARS-CoV-2 Delta variant Switch to treatment with IV Tixa/Cilga 150/150 mg 8 weeks after casirivimab/imdevimab due to prevalence of the SARS-CoV-2 Omicron variant	**Endpoints: SARS-CoV-2 infection, COVID-19-related hospitalization, mortality** 1/6 asymptomatic SARS-CoV-2 infection before Tixa/Cilga No hospitalizations recorded No deaths recorded

Tixa/Cilga; Tixagevimab/Cilgavimab, KTR; kidney transplant recipients, ICU; intensive care unit, RRR; relative risk ratio, HR; hazard ratios, MS; multiple sclerosis, SOTRs; solid organ transplant recipients, BAU: binding antibody units.

## Data Availability

Not applicable.

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
