# Peer review of "Tixagevimab/Cilgavimab in SARS-CoV-2 Prophylaxis and Therapy: A Comprehensive Review of Clinical Experience"

_viruses, 2022, doi:10.3390/v15010118_

Round 1

Reviewer 1 Report

This is a review of the clinical experience of the use of  Tixagevimab/cilgavimab as pre-exposure prophylaxis and treatment for SARS-CoV-2.

The main problems are

-There are no tables or figures to take a quick look at the results (for clinical studies, in vitro monoclonal activity against variants, …), that would facilitate the reading.

In the second version, uploaded to the platform later on, table 1 was added, with a summary of Real World Evidence on effectiveness of Tixagevimab/Cilgavimab in Pre-Exposure Prophylaxis was added. Nonetheless, a table with the activity of monoclonal, or at least Tixagevimab/Cilgavimab will be welcome.

-Statements that are doubtful (ending the pandemic, the inflammatory response, and not the viral replication, is responsible for the severe manifestations, …).

-Absent of comments for the most recent variants (BA.4,6, BQ.1.1) with clear implications in the use of Tixagevimab/cilgavimab and other monoclonals. This is a major drawback of the manuscript. Without this information, the information is incomplete and outdated for clinical use.

-Missing references to escape mutations related to several monoclonals.

a) Abstract

1.- The authors use “ending the pandemic”, and it is clear that the SARS-COV-2 pandemic has not ended. We don’t know if in the future it could be, but now it is very active.

2.- The higher risk of adverse outcomes from SARS-CoV-2 infection didn’t lead to the approval of Tixagevimab/cilgavimab. I understand what you want to mean, but the sentence is not correct.

3.- When talking about prophylaxis use the correct term of pre-exposure or post-exposure prophylaxis for the specific use of Tixagevimab/cilgavimab that it is commented. This combination was not licensed for post-exposure prophylaxis, and this should be made clear in the manuscript.

4.- The loss of in vitro activity of Tixagevimab/cilgavimab against recent variants, like BA.4.6 and BQ.1.1, is not commented on and is a central aspect in the use of this monoclonal combination. BQ.1.1 now is the predominant variant in Europe and in the USA and it is not susceptible in vitro to the Tixagevimab/cilgavimab combination.

b) Introduction

1.- Lines 39-40: “  The latter ( secondary phase of inflammatory response) is responsible for complicated disease, as reflected by severe pneumonia, need for hospitalization, …”.

In the immunocompromised patient, the viral phase is prolonged, and the “inflammatory” phase is less evident. So I would soften the comment.

2.- Missing references to escape mutations related to several monoclonals. This is important, as it is not a problem with one monoclonal, occurs with different monoclonals, and seems particularly frequent in immunocompromised patients.

Some references not included in the manuscript (there are more):

Sotrovimab:

Rockett R, Basile K, Maddocks S, Fong W, Agius JE, Johnson-Mackinnon J, et al. Resistance Mutations in SARS-CoV-2 Delta Variant after Sotrovimab Use. N Engl J Med. 2022;386(15):1477-9.

Gliga S, Luebke N, Killer A, Gruell H, Walker A, Dilthey AT, et al. Rapid selection of sotrovimab escape variants in SARS-CoV-2 Omicron infected immunocompromised patients. Clin Infect Dis. 2022. DOI: 10.1093/cid/ciac802

Birnie E, Biemond JJ, Appelman B, de Bree GJ, Jonges M, Welkers MRA, et al. Development of Resistance-Associated Mutations After Sotrovimab Administration in High-risk Individuals Infected With the SARS-CoV-2 Omicron Variant. JAMA. 2022;328(11):1104-7.

Bamlanivimab + etesevimab

-Pommeret F, Colomba J, Bigenwald C, Laparra A, Bockel S, Bayle A, et al. Bamlanivimab + etesevimab therapy induces SARS-CoV-2 immune escape mutations and secondary clinical deterioration in COVID-19 patients with B-cell malignancies. Ann Oncol. 2021;32(11):1445-7.

Bamlanivimab/etesevimab and casirivimab/imdevimab

-Jary A, Marot S, Faycal A, Leon S, Sayon S, Zafilaza K, et al. Spike Gene Evolution and Immune Escape Mutations in Patients with Mild or Moderate Forms of COVID-19 and Treated with Monoclonal Antibodies Therapies. Viruses. 2022;14(2).

3.- Be specific about what variants you are referring to.

Lines 57-61:  “ …… seem to be inactive against present ones.” Which ones are you referring to?

Lines 61-62: “On the contrary, bebtelovimab and tixagevimab/cilgavimab, remain in vitro active to the circulating subtypes …”.  That is not the case today. For example, BQ.1.1,  the predominant variant in Europe and USA, is not susceptible in vitro to these antibodies.

When talking about Omicron, the subvariant should be specified.

In the FDA fact sheet of EVUSHELD, updated in Nov 2022, the mention to these variants is made:

“VLPs pseudotyped with the SARS-CoV-2 spike of Omicron BQ.1 or BQ.1.1 showed reduced susceptibility to tixagevimab (>1,250-to >10,000-fold) and to cilgavimab (>667-to >5,000-fold).” “VLPs pseudotyped with the spike of Omicron BA.2.12.1, BA.2.75, BA.2.75.2, BA.3, BA.4/BA.5, or BA.4.6 showed 5-fold, 2.4-to 15-fold, >5,000-to >10,000-fold, 16-fold, 33-to 65-fold, or >1,000-fold reductions in neutralizing activity, respectively. VLPs pseudotyped with the spike of Omicron BF.7, BJ.1, BQ.1 or BQ.1.1 showed >5,000-to >10,000-fold, 228-to 424-fold, >2,000-to >10,000-fold or >2,000-to >10,000-fold reductions in neutralizing activity, respectively.”

All this information appears in table 6 of the fact sheet, and for BQ.1, BQ.1.1, BF.7, BA.4.6, BA.2.75.2 with the note: “Tixagevimab and cilgavimab together are unlikely to be active against this variant.”   It is not desirable that in a comprehensive review, the fact sheet gives more updated information.

4.-. The FDA dates of approval and dosage are mentioned. As the journal is an international one, the EMA approvals maybe also interesting for the readers and also to note the different dosages recommended by FDA (300/300 mg) and EMA (150/150) for pre-exposure prophylaxis. In fact, the problem of the different availability of Tixagevimab-cilgavimab is commented on in section six.

5.- Lines 77-78: “The currently recommended dosage in two consecutive injections once every six months, while SARS-CoV-2 remains in circulation”. This is not correct. The dose has to be repeated every 6 months if ongoing protection is needed, and of course, if the SARS-CoV-2 is circulating.

6.- Lines 78-79: “Tixagevimab/ cilgavimab is the only drug indicated for the preexposure prophylaxis of COVID-19 in adults and adolescents.”

It is not correct. In Europe, Ronapreve®  is also indicated for  “Prevention of COVID-19 in adults and adolescents aged 12 years and older weighing at least 40 kg” as pre-exposure prophylaxis or as post-exposure prophylaxis (https://www.ema.europa.eu/documents/product-information/ronapreve-epar-product-information_en.pdf)

So please, specify to which agency or region you are referring to.

7.- No comment is given for antivirals that seem not so variant dependent in their activity compared to monoclonals.

c) Tixagevimab / cilgavimab

1.- Again, specify which type of prophylaxis (pre or post-exposure) you are talking about. Please specify this for all the times you talk about prophylaxis.

2.- Lines 102-103: “The combination is highly potent [18], neutralizing different variants of concern including Omicron”  This is a broad affirmation that is imprecise. Please specify which Omicron variant you are talking about. Several Omicron subvariants are not neutralised by tixagevimab / cilgavimab.

3.- Lines 114-115: “On top of that, regimen has been triply modified (L234F, L235E, P331S) so that Fc effector function is ablated.” Please clarify what you mean by “regimen”.

d) Prophylaxis. Clinical trial data

1.- Lines 147-150: “Among participants who unblinded to receive >1 dose of COVID-19 vaccine (AZD 1222) during the study samples were analyzed for vaccine-induced spike-specific IgG antibodies. No effect of AZD7442 on antibody responses following AZD 1222 COVID-19 vaccination in humans was observed [27].”

I review reference 27 and I was unable to find any comment about AZD7442 and the vaccine response. Please, could you explain where that information appears?

e) Real World Evidence (RWE)

1.-Line 188-190: “Encouragingly, BA.2 neutralization was augmented, and in the current variant climate tixagevimab/cilgavimab pre exposure prophylaxis may serve as a useful complement to vaccination in high-risk SOTRs”

Today, BA.2 is never more a variant that circulates in a significant proportion, so the sentence needs a new redaction.

2.- Specify the type of variant

-Lines 216-218

f) Conclusions

1.- “In conclusion, in the setting of the current pandemic, tixagevimab-cilgavimab represents a useful tool in patients not able to mount an adequate response”

Nowadays, the majority of current variants in USA and Europe (BQ.1.1, BF.7, BA.4.6) are not susceptible in vitro to tixagevimab/cilgavimab.  Therefore, the message needs modification. Remarking the in vitro activity against the circulant variants is essential along with the warning of the increased proportion worldwide of variants not susceptible to tixagevimab-cilgavimab.

g)References

-Nº 11, is incomplete. Add the URL and date of consultation. Moreover, it is more appropriate to quote the FDA fact sheet of the product instead of the company fact sheet.

-Usually, for unpublished publications or pre-prints, the date of online publication and DOI is added. Please add them in reference 28, 29, 31, 32, 33, 36, 39, 40, 41, 43, 45, 46, 48, 49, 55

-URL is missing: 50

Reviewer 2 Report

Akinosoglou, et al summarised Tixagevimab/cilgavimab (AZD7442) in SARS-CoV-2 prophylaxis and therapy. The authors discussed the clinical trial and real-world data against SARS-CoV-2 including BA.1 and BA.2. Overall, this study offered useful information of AZD7442 and may contribute to the control of COVID-19 pandemic.

Major:

(1) It is better to list all included studies as a major result for this comprehensive review. Currently, the manuscript only contained a table including real world evidence of Tixagevimab/Cilgavimab in pre-exposure prophylaxis. More tables should be listed to include all clinical trials or real-world evidence, containing study name, study population, study design, major endpoints, major results, and other critical study information. For example, whether all hospitalization criteria are equivalent should be clarified for table 1.

(2) In the part of Methods, the authors mentioned their searching keywords and study period (2020 Feb 1 - 2022 Nov 15). For a better display of this process, a flowchart describing included and excluded details should be added.

(3) In manuscript, the impact of BA.1 and BA.2 on AZD7442 has been considered, while other circulating strains need to be included.

The circulating strains become other Omicron sub-lineages like BA.4/5 since August 2022. The neutralisation of AZD7442 against BA.4/5 (Tuekprakhon et al., 2022, Cell. DOI:10.1016/j.cell.2022.06.005), BQ.1, and XBB  (Cao, et al., 2022, bioRxiv. DOI: 10.1101/2022.09.15.507787) was reported, which changed NIH The COVID-19 Treatment Guidelines on November 10, 2022. Further analysis of current major strains should be added in the Results, or discussed, or at least mentioned in the Limitations.

Minor:

(1) Since multiple studies have been included, quantitative comparison across studies can be performed. This might help us to illustrate more phenomena. For example, quantitation might help to explain controversy for immunocompromised patients.  If such quantitative analysis cannot be done, some discussion can be added into the manuscript. For example, more discussion on immunocompromised individuals receiving Tixagevimab/cilgavimab (AZD7442) can offer more important opinions.

(2) Line 20: “B-cells depleting” should be “B-cell depleting”.

(3) Line 156: “BA2” should be “BA.2”.

Reviewer 3 Report

The manuscript entitled: "Tixagevimab/cilgavimab in Sars-CoV-2 Prophylaxis and Therapy: A Comprehensive Review of Clinical Experience" by Akinosoglou et al. is a review of the studies and results using this cocktail of antibodies to treat COVID-19 patients or at-risk population, even in a prophylactic and therapeutic form. Either clinical trials or real world evidence was consulted to robust this paper. 

Major comments

Even though is an interesting and useful topic, the lack of the Table in the file (I assume this compiles and shows the data in a better way than the text, it was tough to follow the information in the present form), as well as a deeper discussion in every section, would strengthen the paper, is not clear the relevance of each finding in the overall context.

Minor comments

· Homogenize the way to write the antibodies, i.e. in the title cilgavimab starts with a small letter, then in the introduction is used with capital letters, in the keywords is again with a small letter, so I suggest using a capital letter at the beginning of the name of the antibody throughout the manuscript.

· Please check throughout the manuscript, some times there is a period just before the brackets with the citation, if it is correct (I do not think so), homogenize this in the rest of the text, but if this is an error, anyway needs to be homogenized.

· The correct acronym of the virus is SARS-CoV-2, please correct this in lines 2, 33

·  Line 97, change “COVID-19 virus” by “SARS-CoV-2”.

·  Line 124, does IM stand for intramuscular? Since is the only part that mentions this, it is better to say “intramuscular dose”.

·   Line 129, change ”>60y/o” for “> 60 years old”.

·  Line 138, if ITT is used once in the manuscript, I do not see the usefulness to mention this abbreviation.

·  Line 153, there is no Table along with the manuscript.

· Line 166, a period needs to be changed for a comma after “…COVID-19 vaccine doses”.

· Line 167, change “A reduce in COVID-19 breakthrough infections” to “A reduction in COVID-19 breakthrough infections”.

·  Line 173, homogenize the way to write the name of variants, in most parts of the text start with a small letter, however in this line Omicron was written with a capital letter at the beginning.

· Line 204, homogenize in most parts of the text the cocktail of antibodies is written Tixagevimab/Cilgavimab in this line there is a “hyphen” instead a “slash”.

·  Line 207, a slash is missing in “tixagevimabcilgavimab”.

· Line 210, there is an “a” missing between “…found breakthrough…”

· Line 236, a period after each letter is missing in “eg”, it should be “e.g.”

·  Line 239, whitespace is missing just before the parentheses in “0.75(95%CI,0.58-0.96)” and “0.41(0.19-0.89)”.

·  Line 250-251, the abbreviation RWE should be used instead of “real world evidence”.

· Line 267 whitespaces are missing before and after the number at “age >60years” and a comma is missing after immunosuppression.

· Line 268, the abbreviation ICU should be used instead of “intensive care unit”.  

·  Line 300, a comma is missing after “disease” and a period is missing after “etc”.

· Line 325, define a unique form to write an abbreviation for intramuscular, previously in line 124 IM was used, then the whole word, and now is used “i.m.”

· Line 334, there is an extra parenthesis after “(NCT05234398))” and whitespace is missing after the period.

Round 2

Reviewer 1 Report

The authors have answered satisfactory many of the questions of this reviewer. Nonetheless, there are some issues that require their attention

a) Introduction

1.-Lines 75-76: “ ...and are expected to retain their clinical activity in the future, even though, the duration of their activity remains not well defined in some cases”.

This is a desire, that I can share, but it is speculative. I suggest eliminating it. What we know now is that Tixagevimab/Cilgavimab has no in vitro activity against the most prevalent variants circulating now (BQ.1,  XBB, for example).

b) Table 1

1.-For monoclonals, except Tixagevimab /Cilgavimab, appears in the first row: “no longer recommended”. Please, specify who makes this recommendation (FDA, the authors of the quoted references, the authors of the manuscript, …)

c) Tixagevimab /Cilgavimab

1.-Lines 124-125: “The combination is highly potent [30], neutralizing different variants of concern including Omicron”

This is a broad affirmation that is imprecise and incorrect. Please modify the sentence. Specify which Omicron variant you are talking about. Several Omicron subvariants are not neutralised by tixagevimab / cilgavimab, as previously commented in the manuscript.

2.- In a recent preprint, posted on 7-Nov-2022 (Suribhatla R, et al. Systematic review of the clinical effectiveness of Tixagevimab/Cilgavimab for prophylaxis of COVID-19 in immunocompromised patients. medRxiv. 2022:2022.11.07.22281786) there are 2 studies not included in this review. Please review Suribhatla’s publication and add them if they fulfil the criteria of this review.

3.-Lines 190-195: the study of Chen B et al (reference 57). In this study, the rate of hospitalization in patients that develop COVID after pre-exposure prophylaxis with Tixagevimab/Cilgavimab was significantly lower than in those who developed COVID before receiving  Tixagevimab/Cilgavimab. Please, correct the sentence.

4.-Cardiovascular events

The part dedicated to adverse events, particularly cardiovascular events, is quite brief.

There is a meta-analysis published on 12-Dec-2022 that shows a significant increase in odds of cardiac and vascular adverse events associated with tixagevimab / cilgavimab (Piszczek J, Murthy S, Afra K. Cardiac and vascular serious adverse events following tixagevimab/cilgavimab. The Lancet Respiratory Medicine. DOI: 10.1016/S2213-2600(22)00452-0)

In contrast, a population-based propensity-matched cohort study, published online on 16-Nov-2022, didn't find an association of cardiovascular events with tixagevimab / cilgavimab, and even protection for myocardial infarction (Birabaharan M, Hill E, Begur M, Kaelber DC, Martin TC, Mehta SR. Cardiovascular outcomes after tixagevimab and cilgavimab use for pre-exposure prophylaxis against COVID-19: a population-based propensity-matched cohort study. Clin Infect Dis. 2022. DOI: 10.1093/cid/ciac894)

In the end, the role of tixagevimab / cilgavimab on cardiovascular adverse events remains controversial.

d.-References

1.-Nº 40 is incomplete. I suppose that it refers to a meeting. There was no ECMID meeting in Madrid in 2022. Please add the edition number of the congress, the data and the place.

Author Response

Dear Reviewer#1,

We were very pleased to receive the evaluation of our manuscript and we would like to thank you for your insightful comments. We have addressed the concerns raised in a revised version of the manuscript.

All the new changes in the manuscript are highlighted in blue in the respective file. Please find below our response, point-by-point.

The authors have answered satisfactory many of the questions of this reviewer. Nonetheless, there are some issues that require their attention

 Thank you for your comments, without which the manuscript would not have been improved. All the new changes are highlighted in blue in the respective file/tables.

  1. a) Introduction

1.-Lines 75-76: “ ...and are expected to retain their clinical activity in the future, even though, the duration of their activity remains not well defined in some cases”.

This is a desire, that I can share, but it is speculative. I suggest eliminating it. What we know now is that Tixagevimab/Cilgavimab has no in vitro activity against the most prevalent variants circulating now (BQ.1,  XBB, for example).

The part of the sentence and “are expected to retain their clinical activity in the future” has been eliminated.

  1. b) Table 1

1.-For monoclonals, except Tixagevimab /Cilgavimab, appears in the first row: “no longer recommended”. Please, specify who makes this recommendation (FDA, the authors of the quoted references, the authors of the manuscript, …)

 It has now been corrected. According to NIH living guidance (changes highlighted in blue).

  1. c) Tixagevimab /Cilgavimab

1.-Lines 124-125: “The combination is highly potent [30], neutralizing different variants of concern including Omicron”

This is a broad affirmation that is imprecise and incorrect. Please modify the sentence. Specify which Omicron variant you are talking about. Several Omicron subvariants are not neutralised by tixagevimab / cilgavimab, as previously commented in the manuscript.

 The sentence has been modified. Nonetheless, the authors refer readers to Table 1 for more details.

2.- In a recent preprint, posted on 7-Nov-2022 (Suribhatla R, et al. Systematic review of the clinical effectiveness of Tixagevimab/Cilgavimab for prophylaxis of COVID-19 in immunocompromised patients. medRxiv. 2022:2022.11.07.22281786) there are 2 studies not included in this review. Please review Suribhatla’s publication and add them if they fulfil the criteria of this review.

 Thank you for your comment. Studies included in Suribhalta’s publication and not included in this review are Woopen’s et al case report and a series of patients by Aqeel et al. Authors have now included respective references in text and elaborated in Table 2 (highlighted in blue).

3.-Lines 190-195: the study of Chen B et al (reference 57). In this study, the rate of hospitalization in patients that develop COVID after pre-exposure prophylaxis with Tixagevimab/Cilgavimab was significantly lower than in those who developed COVID before receiving  Tixagevimab/Cilgavimab. Please, correct the sentence.

  Thank you for your comment. It has now been corrected (highlighted in blue)

4.-Cardiovascular events

The part dedicated to adverse events, particularly cardiovascular events, is quite brief.

There is a meta-analysis published on 12-Dec-2022 that shows a significant increase in odds of cardiac and vascular adverse events associated with tixagevimab / cilgavimab (Piszczek J, Murthy S, Afra K. Cardiac and vascular serious adverse events following tixagevimab/cilgavimab. The Lancet Respiratory Medicine. DOI: 10.1016/S2213-2600(22)00452-0)

In contrast, a population-based propensity-matched cohort study, published online on 16-Nov-2022, didn't find an association of cardiovascular events with tixagevimab / cilgavimab, and even protection for myocardial infarction (Birabaharan M, Hill E, Begur M, Kaelber DC, Martin TC, Mehta SR. Cardiovascular outcomes after tixagevimab and cilgavimab use for pre-exposure prophylaxis against COVID-19: a population-based propensity-matched cohort study. Clin Infect Dis. 2022. DOI: 10.1093/cid/ciac894)

In the end, the role of tixagevimab / cilgavimab on cardiovascular adverse events remains controversial.

 Thank you for your comment. The authors have now included and commented the study by Piszczek et al. (highlighted in blue). The references were modified accordingly.

d.-References

1.-Nº 40 is incomplete. I suppose that it refers to a meeting. There was no ECMID meeting in Madrid in 2022. Please add the edition number of the congress, the data and the place.

 Thank you for your comment. You are right it has now been corrected. The references number was modified due to the addition of the study by Piszcek et al.

Reviewer 3 Report

I see a notable updating version, in this, my previous observations were corrected. The addition of table 1 was a good decision. The only comment that I have is the table 2 and the references are in a different font than the rest of the manuscript.

Author Response

Dear Reviewer#3,

We were very pleased to receive the evaluation of our manuscript and we would like to thank you for your insightful comments. We have addressed the concerns raised in a revised version of the manuscript.

I see a notable updating version, in this, my previous observations were corrected. The addition of table 1 was a good decision. The only comment that I have is the table 2 and the references are in a different font than the rest of the manuscript.

Thank you for your comment. References will be formatted following last editing.
